# CL-Calib: Enhancing Post-training Quantization Calibration through Contrastive Learning

## Abstract

Post-training quantization (PTQ) converts a pre-trained full-precision (FP) model into a quantized model in a training-free manner. Determining suitable quantization parameters, such as scaling factors and zero points, is the primary strategy for mitigating the impact of quantization noise (calibration) and restoring the performance of the quantized models. However, the existing activation calibration methods have never considered information degradation between pre- (FP) and post-quantized activations. In this study, we introduce a well-defined distributional metric from information theory, mutual information, into PTQ calibration. We aim to calibrate the quantized activations by maximizing the mutual information between the pre- and post-quantized activations. To realize this goal, we establish a contrastive learning (CL) framework for the calibration, where the quantization parameters are optimized through a self-supervised proxy task. Specifically, by leveraging CL during the PTQ calibration, we can benefit from pulling the positive pairs of quantized and FP activations collected from the same input samples, while pushing negative pairs from different samples. Thanks to the ingeniously designed critic function, we avoid the unwanted but often-encountered collision solution in CL, especially in calibration scenarios where the amount of calibration data is limited. Additionally, we provide a theoretical guarantee that minimizing our designed loss is equivalent to maximizing the desired mutual information. Consequently, the quantized activations retain more information, which ultimately enhances the performance of the quantized network. Experimental results show that our method can effectively serve as an add-on module to existing SoTA PTQ methods.

## 1 Introduction

To meet the growing demand for equipping deep neural networks in resource-constrained edge devices, researchers have developed network quantization techniques [1], in which high-precision parameters and activations are converted into low-precision ones. Quantization methods fall into two main categories: post-training quantization (PTQ) [2, 3, 4, 5, 6] and quantization-aware training (QAT) [7, 8]. QAT involves retraining the model on the labeled training dataset, a process that can be both time-consuming and computationally demanding. In contrast, PTQ only necessitates a small number of unlabeled calibration samples to quantize the pre-trained models, eliminating the need for retraining. This makes PTQ a practical choice for rapid deployment scenarios.

Existing PTQ methods have shown promising results in maintaining good prediction accuracy even when using 4-bit or 2-bit quantization. The majority of these top-performing methods owe their success to meticulous quantization parameter selection. Metrics such as Mean Squared Error (MSE) [9, 5, 10] and cosine distance [11] between the pre- and post-quantization activations in individual layers or modules are commonly used to find the most suitable scaling factors for quantization. These methods are typically referred to as quantized activation calibration techniques. However, existing calibration methods have not taken into account the loss of information during the transition from pre- to post-quantized activation.

In the context of PTQ, we desire the FP and quantized activations (considered as two random variables) to share as much information as possible, as quantized activations are directly

derived from their FP counterparts. Mutual information [12] can measure the amount of information that knowing one of these variables provides about the other. Therefore, in this study, our focus is on the mutual information between FP and quantized activations. Our goal is to optimize PTQ calibration process (particularly, learn the suitable quantization parameters), by maximizing the mutual information between these two types of activations.

By doing so, we aim to preserve as much of the information from the FP activations as possible in the quantized activations, thereby minimizing the loss in model performance due to quantization. Our proposed method is termed "Enhancing PTQ Calibration through Contrastive Learning", abbreviated as CL-Calib. Specifically, we utilize contrastive learning (CT) in the PTQ calibration process. We align positive pairs of quantized and FP activations produced from identical input samples, while simultaneously distancing negative pairs from different samples (see Fig. 1). The collision solution is a common occurrence in CL, often resulting from an insufficiency of samples for each class, and exacerbated by the extreme limitations of calibration data within the PTQ setting. To circumvent this collision, we meticulously design a critic function that leverages the pre-trained full-precision (FP) model. Alongside this straightforward explanation, we also provide a theoretical justification for CL-Calib from the perspective of mutual information. In summary, the quantized activations calibrated by CL-Calib can retain more information, ultimately leading to better performance of the quantized network.

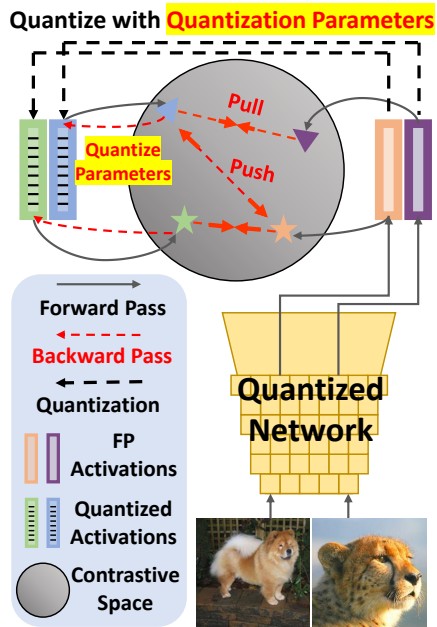

Figure 1: **General Idea: calibrating quantized activation distribution is a contrastive learning manner.** By embedding the activations into a contrastive space, the quantization parameters can be optimized through the pair correlation within the contrastive learning task. *Note that, although this demonstration involves two images, the actual number of images is large.*

Overall, the contributions of this paper are three-fold: **(i)** Under the PTQ calibration framework, we design a self-supervised learning proxy task to optimize the quantization parameters (*e.g.,* , scaling factors) by introducing contrastive learning; **(ii)** We provide a mutual information maximization perspective to understand the quantization problem, and proof that our method can maximize the mutual information between the quantized and full-precision activations; **(iii)** Experimental results demonstrate that our PTQ calibration method, CL-Calib, can function as a plug-and-play module for existing state-of-the-art PTQ methods.

## 2 PRELIMINARY AND BACKGROUND

In this section, we revisit the idea of network quantization and the overarching framework of post-training quantization (PTQ), especially the PTQ activation calibration.

**Basic Notations.** We first define a $K$-layer Multi-Layer Perceptron (MLP). For simplification, we discard the bias term of this MLP. The network $f(\mathbf{X})$ can be denoted as:

$$f(\mathbf{W}^1, \cdots, \mathbf{W}^K; \mathbf{X}) = (\mathbf{W}^K \cdot \sigma \cdot \mathbf{W}^{K-1} \cdot \cdots \cdot \sigma \cdot \mathbf{W}^1)(\mathbf{X}), \tag{1}$$

where $\mathbf{X}$ is the input sample and $\mathbf{W}^k : \mathbb{R}^{d_{k-1}} \longmapsto \mathbb{R}^{d_k} (k = 1, ..., K)$ stands for the weight matrix connecting the $(k-1)$-th and the $k$-th activation layer, with $d_{k-1}$ and $d_k$ representing the sizes of the input and output of the $k$-th network layer, respectively. We denote $f^k(\mathbf{W}^k; \cdot)$ as the $k$-th layer's mapping function. The $\sigma(\cdot)$ function performs element-wise activation operations on the input feature maps. Utilizing the aforementioned predefined notions, we define a *sliced MLP* $f^{[:k]}(\mathbf{x})$ and $f^{[k:]}(\mathbf{x})$, which consists of the first $k$ layers and the last $(K - k)$ layers of $f(\mathbf{x})$, as follows:

$$
\begin{aligned}
\mathbf{A}^k = f^{[:k]}(\mathbf{W}^1, \cdots, \mathbf{W}^k; \mathbf{X}) &= (\mathbf{W}^k \cdot \sigma \cdots \sigma \cdot \mathbf{W}^1)(\mathbf{X}), \\
f^{[k:]}(\mathbf{W}^k, \cdots, \mathbf{W}^K; \mathbf{A}^k) &= (\mathbf{W}^K \cdot \sigma \cdots \sigma \cdot \mathbf{W}^k)(\mathbf{A}^k),
\end{aligned}
\tag{2}
$$

where $\mathbf{A}^k$ is the $k$-th layer's output activation. And the MLP $f$ can be seen as a special case in the sliced function sequences $\{f^{[:k]}\}$ and $\{f^{[k:]}\}$ ($k \in \{1, \cdots, K\}$), *i.e.*, , $f = f^{[:K]} = f^{[0:]}$. For ease of reference, we use subscripts $F$ and $Q$ for $f$ and $\mathbf{A}$ to represent the full-precision and quantized versions, respectively.

**Post-training quantization (PTQ)** takes a well-trained full-precision network $f_F$ as input and selects quantization parameters to quantize the weight tensor and activation tensor in each layer for obtaining quantized network $f_Q$. *To convert a tensor into a quantized tensor, only two quantization parameters are required, i.e., the scaling factor $S$ and the zero point $Z$.* Consequently, most PTQ methods primarily focus on selecting appropriate quantization parameters [2, 3, 4, 5, 6]. One of the most prevalent approaches for selecting parameters is to minimize the error induced by quantization. The quantization process can be formulated as an optimization problem:

$$\arg\min_{S,Z} \mathcal{L}_{quant}, \text{ where } \mathcal{L}_{quant} = \text{Metric}(\mathbf{X}_Q, \mathbf{X}_F), \tag{3}$$

where Metric is the metric function measuring the distance between $\mathbf{X}_Q$ and the full-precision tensor $\mathbf{X}_F$. MSE, cosine distance, L1 distance, and KL divergence are commonly used metric functions. Formally, after obtaining appropriate $S$ and $Z$, a full-precision tensor can be converted, as follows:

$$\mathbf{X}_Q = S(\text{clamp}(\lfloor \frac{\mathbf{X}_F}{S} \rceil - Z, p_{min}, p_{max}) + Z), \tag{4}$$

where $[p_{min}, p_{max}]$ is the quantization range determined by bit-width, for 8 bit integer, the range is $[-128, 127]$. Note that, in this work We only consider uniform unsigned symmetric quantization, as it is the most widely used quantization setup, and $\mathbf{X}_Q$ is not in an integer form, but is dequantized float value ($\mathbf{X}_Q \approx \mathbf{X}_F$ is easily for alignment in Eq. 3, while $\frac{\mathbf{X}_Q}{S}$ is in integer form).

**Activation Calibration.** Although the network's weights can be quantized without data by minimizing the quantization error (Eqs. 3 and 4), a similar approach cannot be simply used for activation quantization, as the activation tensors are inaccessible without input. To collect the activation tensors, a set of unlabeled input samples (calibration dataset) is used as the network input. The size of the calibration dataset (*e.g.,* , 128 randomly selected images in ImageNet [13]) is significantly smaller than the training dataset. After obtaining the $k$-th layer's full-precision activation $\mathbf{A}_F^k$, the quantized activation $\mathbf{A}_Q^k$ can be similarly calibrated based on Eqs. 4 and 3.

In general, PTQ quantizes a network in three steps: **(i)** Select which operations in the network should be quantized and leave the other operations in full-precision. For example, some special functions such as softmax and GeLU often takes full-precision [14]. **(ii)** Collect the calibration samples. The distribution of the calibration samples should be as close as possible to the distribution of the real data to avoid over-fitting of quantization parameters on calibration samples. **(iii)** Use the proper method to select quantization parameters for weight and activation tensors. *Recently, state-of-the-art PTQ works [6, 5, 10] reveal that the bottleneck for further improving the PTQ performance is the activation quantization rather than weight quantization.* They focus on **activation calibration**. Specifically, determining the quantization hyper-parameters (*i.e.*, scaling factor $S_a^k$ and zero point $Z_a^k$) for quantizing the $k$-th layer's activation is addressed as an optimization problem:

$$\arg\min_{S_a^k, Z_a^k} \mathcal{L}_{calib}, \text{ where } \mathcal{L}_{calib} = \text{Metric}(\mathbf{A}_Q^k, \mathbf{A}_F^k), \tag{5}$$

in which $\mathcal{L}_{calib}$ is the calibration objective for activation quantiztaion. In this paper, we also focus on enhancing the activation calibration (step **(iii)**). Specifically, we introduce contrastive learning as a proxy task during the calibration phase. By leveraging the proxy task, the quantization parameters are optimized based on self-supervised signals (Sec. 3.1), making the quantized activations more similar to their full-precision counterparts in terms of mutual information (Sec. 3.2).

## 3 METHOD

In this section, we elucidate the methodology for PTQ calibration via contrastive learning, which we refer to as `CL-Calib`. On the one hand, we straightforwardly demonstrate the construction of a self-supervised learning proxy task within the context of the PTQ calibration framework, as well as the associated optimization formulation. On the other hand, based on the method, we give a theoretical explanation for the efficacy of `CL-Calib` through the lens of mutual information maximization.

### 3.1 METHODOLOGY: CL-CALIB

**Instance Recognition (IR) for PTQ Calibration.** As pioneers, [15, 16] propose instance recognition as a self-supervised learning task, categorizing images based on their unique labels. Specifically, for a image $\mathbf{X}$ in the dataset $\mathcal{D}$ following $p_{data}$, a learnable encoder $g_\theta$ mapping the image $\mathbf{X}$ to a feature vector $\mathbf{V} = g(\mathbf{X})$. The resulting vector $\mathbf{V}$ should serve as an accurate representation of $\mathbf{X}$. To accomplish this task, contrastive learning (CL) approaches use a training method that distinguishes a positive from multiple negatives, based on the similarity principle between samples as shown in Fig. 2. The InfoNCE [17, 16] loss function, a popular choice for CL, can be expressed as:

$$\mathcal{L}_{contrstive} = \mathbb{E}_{(\mathbf{X},\mathbf{X}_+)\sim p_{\text{pos}}, \{\mathbf{X}_{-,i}\}_{i=1}^{M} \overset{\text{i.i.d}}{\sim} p_{\text{data}}} \left[ -\log \frac{\exp(\mathbf{V} \cdot \mathbf{V}_+/\tau)}{\exp(\mathbf{V} \cdot \mathbf{V}_+/\tau) + \sum_{i=1}^{M} \exp(\mathbf{V} \cdot \mathbf{V}_{-,i}/\tau)} \right\} \right],$$
(6)

where $\mathbf{X}_+$ represents a positive sample for $\mathbf{X}$, and $\mathbf{X}_{-,i}$ represents the $i$-th negative sample for $\mathbf{X}$. The symbol "·" refers to the inner (dot) product, and $\tau > 0$ is a temperature hyper-parameter. It is worth noting that the embeddings used in the loss function are normalized by L2 normalization.

From a representation learning perspective, we can interpret the aforementioned loss function in an intuitive manner. The term for positive pairs is optimized to accentuate intra-class correlations, while the term for negative pairs serves to promote inter-class decorrelation. Because pairs are constructed instance-wisely, the number of negative samples can, theoretically, be as large as the size of the entire training set. Several CL methods [18, 16, 19, 20] have demonstrated the benefits of enhancing the similarity between representations in multiple views as shown in Fig. 2. These methods inspire that by enhancing the correlation between quantized and full-precision (FP) activations, we can improve the representational capacity of quantized networks, thereby improving the performance of the quantized model.

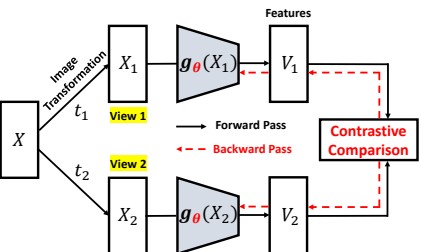

Figure 2: In **contrastive instance learning**, the features produced by different transformations of the same sample are contrasted to each other.

**For the calibration of the $k$-th layer's quantized activation $\mathbf{A}_Q^k$**, only two quantization parameters are required: scaling factor $S_a^k$ and zero point $Z_a^k$ are required. After obtaining suitable quantization parameters, $\mathbf{A}_Q^k$ can be quantized as follows:

$$\mathbf{A}_Q^k = S_a^k(\text{clamp}(\lfloor \frac{\mathbf{A}_F^k}{S_a^k} \rceil - Z_a^k, p_{min}, p_{max}) + Z_a^k).$$
(7)

The essence of contrastive learning (CL) is to compare different views of the data (typically under various data augmentations) for calculating similarity scores. This CL approach is effective in *defining the view and negative samples* in a manner that can be numerically accessed [16, 19, 20, 21, 18]. Therefore, to learn $S_a^k$ and $Z_a^k$ via CL, we must first define the views, and corresponding positive and negative pairs for comparison. Generally, we collect positive pairs by the quantized and FP activations from the same input sample, while negative pairs are collected from different samples. By optimizing the CL task, we are able to learn the quantization parameters that make the quantized activations contain more information, and hence, more similar to the FP ones in terms of information. This can result in a more effective PTQ calibration.

**Definitions: Views and Negative Samples for CL-Calib.** Notably, during the PTQ calibration phase, we naturally have two views of the data: the quantized activations $A_Q$ and the FP activations $A_F$. We leverage these two views, together with additional contrastive pairs, to improve the correlation between the quantized and full-precision activations (see Eq. 7 and Fig. 3). Specifically, For a calibration batch with $M + 1$ samples, the samples can be denoted as: $\{\mathbf{X}_i\}(i \in \{1, \cdots, M + 1\})$. We feed a batch of samples to the quantized network $f_Q^{[:k]}$ and obtain $(M + 1)^2$ pairs of activations $(\mathbf{A}_{Q,i}^k, \mathbf{A}_{F,j}^k)$, which augments the data for the auxiliary task. We define a pair containing two activations from the same sample as positive pair, *i.e.*, , if $i = j$, $(\mathbf{A}_{Q,i}^k, \mathbf{A}_{F,+,j}^k)$ and *vice versa* $(\mathbf{A}_{Q,i}^k, \mathbf{A}_{F,-,j}^k)$. The core idea of contrastive learning is to discriminate whether a given pair of activation $(\mathbf{A}_{Q,i}^k, \mathbf{A}_{F,j}^k)$ is positive or negative with a learnable neural network $d(\cdot, \cdot)$, *i.e.*, , estimating

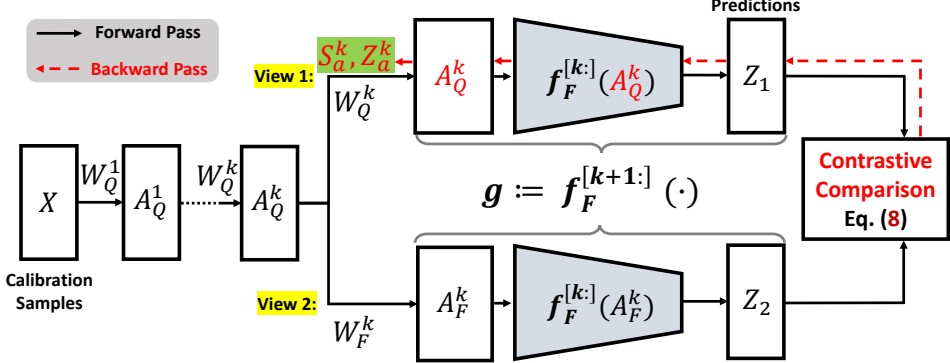

Figure 3: **Calibrating the activation of k-th layer.** The goal is to optimize the quantization parameters (*i.e.,* , scaling factor $S_a^k$ and zero point $Z_a^k$) for quantizing the activation of k-th layer. Feeding a set of calibration images into a quantized network, we can obtain the full-precision and quantized activations (*i.e.,* , $A_F^k$ and $A_Q^k$). Then, we embed the activations into a prediction space via $g = f_F^{[k:]}$ (with frozen FP parameters). By optimizing the pair correlation within the contrastive learning task in Eq. 8, we can refine the scaling factor for calibrating the activation of the $k$-th layer.

the distribution $P(D \mid \mathbf{A}_{Q,i}^k, \mathbf{A}_{F,j}^k)$ with $d$, in which $D$ is the latent variable determining whether $i = j$ or $i \neq j$. The $d$ has a form as $d(\mathbf{A}_{Q,i}^k, \mathbf{A}_{F,j}^k) \triangleq \frac{\exp(g(\mathbf{A}_{Q,i}^k) \cdot g(\mathbf{A}_{F,j}^k)/\tau)}{\exp(g(\mathbf{A}_{Q,i}^k) \cdot g(\mathbf{A}_{F,j}^k)/\tau)+1}$. Upon setting up all the prerequisites for CL during the calibration phase, we can define the CL-Calib loss as follows:

$$\mathcal{L}_{\texttt{CL-Calib}} = \mathbb{E}_{(\mathbf{A}_Q^k, \mathbf{A}_{F,+}^k)} \left[ -\log d(\mathbf{A}_Q^k, \mathbf{A}_{F,+}^k) \right] + \mathbb{E}_{\{\mathbf{A}_{F,-,i}^k\}_{i=1}^M} \left[ -\log(1 - d(\mathbf{A}_Q^k, \mathbf{A}_{F,-,i}^k)) \right], \quad (8)$$

in which $\mathbf{A}_Q^k$ is the $k$-th layer's quantized output, *i.e.,* , $\mathbf{A}_Q^k = f_Q^{[:k]}(\mathbf{W}_Q^1, \cdots, \mathbf{W}_Q^k; \mathbf{X})$. Theoretically, minimizing $\mathcal{L}_{\texttt{CL-Calib}}$ is able to produce a binary discriminator $d^\star$, which can classify a given pair $(\mathbf{A}_{Q,i}^k, \mathbf{A}_{F,j}^k)$ into positive or negative.

Combining the designed CL-Calib loss, the overall calibration objective $\mathcal{L}_{calib}$ can be defined as:

$$\mathcal{L}_{calib} = \mathcal{L}_{quant} + \lambda \mathcal{L}_{\texttt{CL-Calib}}, \quad (9)$$

where $\mathcal{L}_{quant}$ is the activation reconstruction loss (*e.g.,* , state-of-the-arts QDrop [10] and PD-Quant [6] in practice), $\lambda$ is used to control the degree of NCE loss. Straightforwardly, the quantized and FP activations from identical samples can be pulled close, and activations from different samples can be pushed away, which corresponds

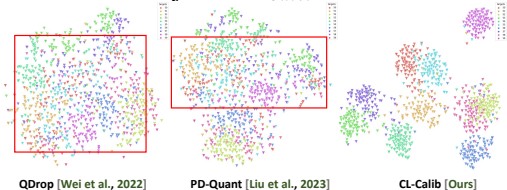

Figure 4: t-SNE visualization for activations calibrated by different calibration methods.

to the core idea of contrastive learning. Hence, the representation ability of the quantized activations calibrated by our method is enhanced, as demonstrated in Fig. 4.

**Network Architecture of mapping function $g$.** Because there is only one instance per class in contrastive instance discrimination, training stability is significantly compromised, resulting in considerable optimization fluctuations. This instability often leads CL algorithms towards collision solutions [18, 16, 19, 20]. In the context of PTQ calibration, the limited availability of calibration data further exacerbates the optimization challenge. We carefully design the mapping network $g$ by leveraging the frozen FP network $f_F^{[k:]}$ as shown in Fig. 3. Specifically, instead of training a fully-learnable network from scratch as most CL methods do, we utilize the existing pre-trained model $f_F^{[k:]}$ as the mapping network. The benefits of using $f_F^{[k:]}$ are threefold: **(i)** $f_F^{[k:]}$ represents a well-trained mapping from $\mathbf{A}_{F,j}^k$ to the prediction, corresponding to the task itself. Therefore, the need to design a contrastive space is eliminated. Empirically, we have also attempted to use a fully-learnable network as $g$. However, its optimization fails, mainly due to the limited training data available for traditional CL. **(ii)** The primary goal is to optimize the scaling factor and zero point for the quantization of the $k$-th layer's activation (involving only two parameters). As such, learning $g$

proves to be a thankless task. **(iii)** Foregoing the learning of $g$ is in line with the application scenarios of PTQ, that is, rapid network deployment.

## 3.2 THEORETICAL GUARANTEE: MUTUAL INFORMATION MAXIMIZATION

In this section, we provide a theoretical explanation as to why `CL-Calib` is capable of maximizing the mutual information during the quantization process, *i.e.,* , maximizing the mutual information between full-precision and quantized activations for calibration.

**Mutual Information and Contrastive Learning.** For two discrete variables $\mathbf{X}$ and $\mathbf{Y}$, their mutual information (MI) can be defined as [12]:

$$I(\mathbf{X}, \mathbf{Y}) = \sum_{x,y} P_{\mathbf{XY}}(x, y) \log \frac{P_{\mathbf{XY}}(x, y)}{P_{\mathbf{X}}(x) P_{\mathbf{Y}}(y)}, \quad (10)$$

where $P_{\mathbf{XY}}(x, y)$ is the joint distribution, $P_{\mathbf{X}}(x) = \sum_y P_{\mathbf{XY}}(x, y)$ and $P_{\mathbf{Y}}(y) = \sum_x P_{\mathbf{XY}}(x, y)$ are the marginals of $\mathbf{X}$ and $\mathbf{Y}$, respectively. Mutual information quantifies the amount of information gained about one random variable through observing another random variable. It is a dimensionless quantity, typically measured in bits, and can be regarded as the reduction in uncertainty about one random variable given knowledge of the other. High mutual information signifies a substantial reduction in uncertainty, and *vice versa* [12].

Back to the content of calibrating the $k$-th layer's quantizing activation via CL, activations $\mathbf{A}_F^k$ and $\mathbf{A}_Q^k$ can be considered as two variables. The corresponding variables should share more information, *i.e.,* , the mutual information of the activations $I(\mathbf{A}_F^k, \mathbf{A}_Q^k)$ should be maximized to enforce them mutually dependent. The core idea of CL is to discriminate whether a given pair of activation $(\mathbf{A}_{Q,i}^k, \mathbf{A}_{F,j}^k)$ is positive or negative, *i.e.,* , inferring the distribution $P(D \mid \mathbf{A}_{Q,i}^k, \mathbf{A}_{F,j}^k)$, in which $D$ is the variable decides whether $i = j$ or $i \neq j$. However, accessing $P(D \mid \mathbf{A}_{Q,i}^k, \mathbf{A}_{F,j}^k)$ directly is not feasible [17], so we introduce its variational approximation instead

$$q(D \mid \mathbf{A}_{Q,i}^k, \mathbf{A}_{F,j}^k), \quad (11)$$

which can be estimated by our models $d(\cdot, \cdot)$. Intuitively, $q(D \mid \mathbf{A}_{Q,i}^k, \mathbf{A}_{F,j}^k)$ can be treated as a binary classifier, which can classify a given pair $(\mathbf{A}_{Q,i}^k, \mathbf{A}_{F,j}^k)$ into positive or negative. From Bayes' theorem, we can formalize the posterior probability of two activations coming from a positive pair as follows:

$$q(D = 1 \mid \mathbf{A}_{Q,i}^k, \mathbf{A}_{F,j}^k) = \frac{q(\mathbf{A}_{Q,i}^k, \mathbf{A}_{F,j}^k \mid D = 1)\frac{1}{M+1}}{q(\mathbf{A}_{Q,i}^k, \mathbf{A}_{F,j}^k \mid D = 1)\frac{1}{M+1} + q(\mathbf{A}_{Q,i}^k, \mathbf{A}_{F,j}^k \mid D = 0)\frac{M}{M+1}}. \quad (12)$$

The probability of activations from negative pair is $q(D = 0 \mid \mathbf{A}_{Q,i}^k, \mathbf{A}_{F,j}^k) = 1 - q(D = 1 \mid \mathbf{A}_{Q,i}^k, \mathbf{A}_{F,j}^k)$. To simplify the NCE derivative, several works [17, 15, 22, 23] build assumptions about the dependence of the variables, we also use the assumption that the activations from positive pairs are dependent and the ones from negative pairs are independent, *i.e.* $q(\mathbf{A}_{Q,i}^k, \mathbf{A}_{F,j}^k \mid D = 1) = P(\mathbf{A}_{Q,i}^k, \mathbf{A}_{F,j}^k)$ and $q(\mathbf{A}_{Q,i}^k, \mathbf{A}_{F,j}^k \mid D = 0) = P(\mathbf{A}_{Q,i}^k)P(\mathbf{A}_{F,j}^k)$. Hence, the above equation can be simplified as:

$$q(D = 1 \mid \mathbf{A}_{Q,i}^k, \mathbf{A}_{F,j}^k) = \frac{P(\mathbf{A}_{Q,i}^k, \mathbf{A}_{F,j}^k)}{P(\mathbf{A}_{Q,i}^k, \mathbf{A}_{F,j}^k) + P(\mathbf{A}_{Q,i}^k)P(\mathbf{A}_{F,j}^k)M}. \quad (13)$$

By applying the logarithm to Eq. 13 and rearranging the terms, we obtain the following equation:

$$\log q(D = 1 \mid \mathbf{A}_{Q,i}^k, \mathbf{A}_{F,j}^k) \leq \log \frac{P(\mathbf{A}_{Q,i}^k, \mathbf{A}_{F,j}^k)}{P(\mathbf{A}_{Q,i}^k)P(\mathbf{A}_{F,j}^k)} - \log(M). \quad (14)$$

By taking expectation on both sides with respect to $P(\mathbf{A}_{Q,i}^k, \mathbf{A}_{F,j}^k)$, and combining the definition of mutual information in Eq. 10, we can derive the form of mutual information as:

$$\overbrace{I(\mathbf{A}_Q^k, \mathbf{A}_F^k)}^{\text{targeted } \mathbf{MI}} \geq \overbrace{\mathbb{E}_{P(\mathbf{A}_{Q,i}^k, \mathbf{A}_{F,j}^k \mid D=1)} \left[ \log q(D = 1 \mid \mathbf{A}_{Q,i}^k, \mathbf{A}_{F,j}^k) \right] + \log(M)}^{\text{lower bound}}, \quad (15)$$

where $I(\mathbf{A}_Q^k, \mathbf{A}_F^k)$ is the mutual information between the quantized and full-precision distributions of our targeted object. Instead of directly maximizing the mutual information, maximizing the lower bound in the Eq. 15 is a practical solution. We can further loosen the lower bound, and get the optimization objective corresponding to CL-Calib loss in Eq. 8 as follows:

$$I(\mathbf{A}_Q^k, \mathbf{A}_F^k) - \log(M)$$
$$\geq \mathbb{E}_{P(\mathbf{A}_{Q,i}^k, \mathbf{A}_{F,j}^k | D=1)} \left[ \log q(D=1 \mid \mathbf{A}_{Q,i}^k, \mathbf{A}_{F,j}^k) \right] + M\mathbb{E}_{P(\mathbf{A}_{Q,i}^k, \mathbf{A}_{F,j}^k | D=0)} \left[ \log q(D=0 \mid \mathbf{A}_{Q,i}^k, \mathbf{A}_{F,j}^k) \right]$$
$$\underbrace{\phantom{xxxxxxxxxxxxxxxxxxxxxxxxxxxxxxxxxxxxxxxxxxxxxxxxxxxxxxxxxxxxxxxxxxxxxxxxxxxxxxxxxxxxx}}_{\text{negative CL-Calib loss in Eq. 8}}$$
$$= \mathbb{E}_{\mathbf{A}_Q^k}\{\overbrace{\mathbb{E}_{(\mathbf{A}_Q^k, \mathbf{A}_{F,+}^k)} \left[ \log d(\mathbf{A}_Q^k, \mathbf{A}_{F,+}^k) \right] + \mathbb{E}_{\{\mathbf{A}_{F,-,i}^k\}_{i=1}^M} \left[ -\log(1 - d(\mathbf{A}_Q^k, \mathbf{A}_{F,-,i}^k)) \right]}\}.$$

$$(16)$$

By minimizing the overall calibration objective $\mathcal{L}_{calib}$ in Eq. 9, we can obtain the function $d^\star$. This function serves as a variational approximation of the distribution $P(D = 1 \mid \mathbf{A}_Q^k, \mathbf{A}_F^k)$. In this way, we can optimize the lower bound of targeted mutual information $I(\mathbf{A}_Q^k, \mathbf{A}_F^k)$. This validates our assertion that employing CL-Calib to learn activation quantization parameters effectively maximizes the mutual information between pre- and post-quantized activations.

## 4 EXPERIMENTS

In this section, we perform experiments on image classification with ImageNet dataset [13] to validate the effectiveness of CL-Calib. Furthermore, we designate a series of ablative and analytical studies to verify the effectiveness and investigate various properties of CL-Calib including the regularization, and reliability. All experiments are implemented using PyTorch [24] on 8 Nvidia RTX A6000 and **codes** are in the supplementary.

Table 1: Comparison of CL-Calib with various post-training quantization algorithms on ImageNet.

| Methods | Bits (W/A) | ResNet-18 | ResNet-50 | MobileNetV2 | RegNetX-600MF | RegNetX-3.2GF | MNasx2 |
|---|---|---|---|---|---|---|---|
| Full Prec. | 32/32 | 71.01 | 76.63 | 72.62 | 73.52 | 78.46 | 76.52 |
| ACIQ-Mix [25] | | 67.00 | 73.80 | - | - | - | - |
| LAPQ [26] | | 60.30 | 70.00 | 49.70 | 57.71 | 55.89 | 65.32 |
| Bit-Split [27] | | 67.00 | 73.80 | - | - | - | - |
| AdaRound [3] | 4/4 | 67.96 | 73.88 | 61.52 | 68.20 | 73.85 | 68.86 |
| QDrop [10] | | 69.17 | 75.15 | 68.07 | 70.91 | 76.40 | 72.81 |
| PD-Quant [6] | | 69.30 | 75.09 | 68.33 | 71.04 | **76.57** | 73.30 |
| CL-Calib | | **69.41** (+0.11) | **75.38** (+0.23) | **68.56** (+0.23) | **71.38** (+0.34) | 76.40 (-0.17) | **73.60** (+0.30) |
| LAPQ [26] | | 0.18 | 0.14 | 0.13 | 0.17 | 0.12 | 0.18 |
| Adaround [3] | | 0.11 | 0.12 | 0.15 | - | - | - |
| QDrop [10] | 2/4 | 64.57 | 70.09 | 53.37 | 63.18 | 71.96 | 63.23 |
| PD-Quant [6] | | 65.07 | **70.92** | 55.27 | 64.00 | 72.43 | 63.33 |
| CL-Calib | | **65.14** (+0.07) | **70.92** (±0.00) | **55.63** (+0.36) | **64.50** (+0.50) | **72.82** (+0.39) | **63.46** (+0.13) |
| QDrop [10] | | 57.56 | 63.26 | 17.30 | 49.73 | 62.00 | 34.12 |
| PD-Quant [6] | 4/2 | 58.65 | 64.18 | 20.40 | 51.29 | 62.76 | 38.89 |
| CL-Calib | | **59.03** (+0.38) | **65.12** (+0.96) | **22.77** (+2.37) | **52.35** (+1.06) | **63.53** (+0.77) | **40.80** (+1.91) |
| BRECQ [5] | | 42.54 | 29.01 | 0.24 | 3.58 | 3.62 | 0.61 |
| AdaQuant [28] | | 0.11 | 0.12 | 0.15 | - | - | - |
| QDrop [10] | 2/2 | 51.42 | 55.45 | 10.28 | 39.01 | 54.38 | 23.59 |
| PD-Quant [6] | | 53.08 | 56.98 | 14.17 | 40.92 | 55.13 | 28.03 |
| CL-Calib | | **54.45** (+1.37) | **58.30** (+1.32) | **17.70** (+3.53) | **42.19** (+1.27) | **56.39** (+1.26) | **30.34** (+2.31) |

### 4.1 EXPERIMENT SETTINGS

To evaluate our proposed method, we quantize various Convolutional Neural Network (CNN) architectures, including ResNet [29], MobileNetV2 [30], RegNet [31], and MnasNet [32]. The full-precision (FP) pre-trained models used in our experiments are sourced from [5]. We evaluate CL-Calib on the ImageNet dataset, using a batch size of 128. For calibration, we randomly sample 128 images from the ImageNet training dataset. Unless specified otherwise, we set the first and last layer quantization to 8-bit for all PTQ experiments. Furthermore, we maintain the same quantization settings and hyper-parameters as used in the QDrop [10] and PD-quant [6] (previous state-of-the-art methods) implementations. The learning rate for the activation quantization scaling factor is set to 4e-5, and for weight quantization rounding, the learning rate is set to 3e-3. The choice of hyper-parameters $\tau$ and $\lambda$ in Eq. 8 and Eq. 9 will be discussed in Sec. 4.3, respectively. The fine-tuning of quantization parameters is performed over 20,000 iterations.

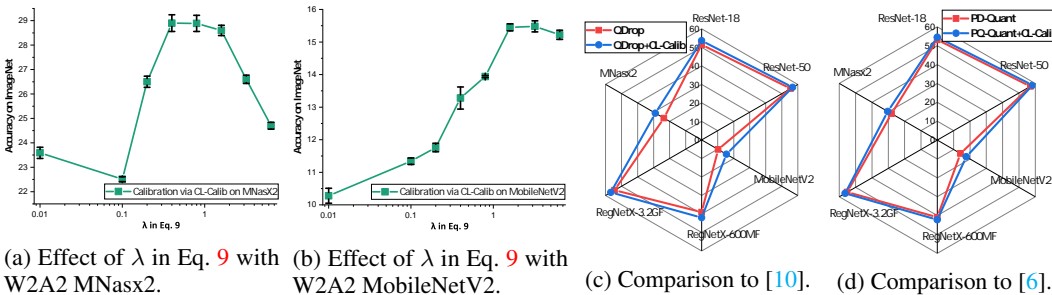

(a) Effect of $\lambda$ in Eq. 9 with W2A2 MNasx2.
(b) Effect of $\lambda$ in Eq. 9 with W2A2 MobileNetV2.
(c) Comparison to [10].
(d) Comparison to [6].

Figure 5: **Ablation Studies & Performance Summary.**

## 4.2 COMPARISON TO STATE-OF-THE-ARTS

In a thorough comparison of our method `CL-Calib` with numerous PTQ algorithms across different bit settings, we observe that `CL-Calib` consistently surpasses state-of-the-art PTQ methods, notably in extremely low-bit scenarios. Notably, merely optimizing the activation scaling factors does not provide satisfactory results at low bits; therefore, unless otherwise stated, all our subsequent experiments optimize both the rounding values and activation scaling factors, following the precedent set by previous SoTAs such as QDrop and PD-Quant [10, 6]. The outcomes are summarized in Tab. 1, demonstrating significant improvements achieved by `CL-Calib` compared to the previously SoTA PTQ baselines. For instance, when quantizing the network to W4A4, experiments indicate that `CL-Calib` slightly surpasses PD-Quant. However, as the bit-width decreases, the advantages of `CL-Calib` become increasingly apparent. At the most challenging W2A2 bit setting, `CL-Calib` exceeds the baseline across all network architectures, in particular, `CL-Calib` boosts the accuracy of post-quantized MobileNetV2 by 3.43% and MNasx2 by 2.31%.

There are more extensive experiments in the Appendix with different architectures, including the **Vision Transformer** [33] (Tab. 3 in Appendix), and on various tasks, such as **object detection** (Tab. 5 in Appendix). We can witness the generalization ability of `CL-Calib` from those experiments.

Table 2: Serving as an add-on module on ImageNet with 2W2A setting.

| Methods | ResNet-18 | ResNet-50 | MobileNetV2 | RegNetX-600MF | RegNetX-3.2GF | MNasx2 |
|---|---|---|---|---|---|---|
| Full Prec. | 71.01 | 76.63 | 72.62 | 73.52 | 78.46 | 76.52 |
| PD-Quant | 53.08 | 56.98 | 14.17 | 40.92 | 55.13 | 28.03 |
| CL-Calib | **54.45** (+1.37) | **58.30** (+1.32) | **17.70** (+3.53) | **42.19** (+1.27) | **56.39** (+1.26) | **30.34** (+2.31) |
| QDrop | 51.42 | 55.45 | 10.28 | 39.01 | 54.38 | 23.59 |
| CL-Calib | **53.63** (+3.21) | **56.78** (+1.33) | **15.48** (+4.20) | **42.00** (+0.99) | **56.65** (+2.27) | **28.89** (+5.30) |

## 4.3 FURTHER ANALYSIS

**Ablative Studies.** We conduct a series of ablation studies of `CL-Calib` on the most challenging W2A2 settings with MobileNetV2 and MNasx2 architectures. By adjusting the coefficient $\lambda$ in the loss function $\mathcal{L}_{calib}$ (Eq.9), we examine the influence of `CL-Calib` loss in calibration. The results, illustrated in Figs.5c and 5d, show a trend where increasing $\lambda$ improves the performance, thus validating the effectiveness of our proposed method. In all our experiments, we use the PQ-Quant [6] codebase. Setting $\lambda = 0$ is our baseline, which corresponds to PD-Quant.

**Serving as an Add-on Module.** We implement `CL-Calib` on the SoTA PTQ methods, such as QDrop [10], and PD-Quant [6] with extreme W2A2 PTQ setting. The results are presented in Fig. 5a,5b, and Tab. 2. We can witness that `CL-Calib` can improve the performances of all the methods. This is a piece of evidence that our method can serve as an add-on module.

**Hyper-parameters and Relative Module Selection.** In addition to ablation studies, we conduct experiments to select the important hyper-parameters and modules. We investigate two hyper-parameter, the co-efficient $\tau$ to adjust the temperature in the `CL-Calib` loss in Eq. 8, the number of negative samples $M$, the architecture of embedding network $g$ in critic function, and the similarity metric for contrastive learning. The results are in the Appendix (A.4).

**Representation Ability**. We examined the representation capability of the quantized activations through t-SNE visualization [34]. The results, shown in Fig. 4, indicate that the activations calibrated by `CL-Calib` exhibit superior representation capacity.

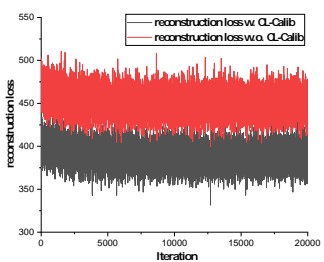

**Regularization: Mitigating Overfitting.** We scrutinize the evolution of the quantization loss, $\mathcal{L}_{quant}$, as given in Eq.8 and Eq.9 during training. The term $\mathcal{L}_{quant}$ in Eq.9 can represent any quantization loss, which *is easily overfitted due to the limited calibration data*. By manipulating the coefficient $\lambda$ in Eq.9, we can examine $\mathcal{L}_{quant}$ in isolation. The training curves are depicted in Fig.6. Upon enabling the CL-Calib loss $\mathcal{L}_{CL-Calib}$ (by setting $\lambda = 3.2$), we notice a reduction in $\mathcal{L}_{quant}$ compared to when our designed CL-Calib is not used (also see Tab. 7 in Appendix). This decrease is accompanied by an improvement in the corresponding test performance (see Tab. 2). Hence, we can deduce that CL-Calib module functions as a regularization term.

Figure 6: Training curves of quantization loss $\mathcal{L}_{quant}$.

## 5 RELATED WORK

**Quantization** stands as one of the most potent techniques for compressing neural networks. It can be broadly categorized into two primary methods: Quantization-Aware Training (QAT) and Post-Training Quantization (PTQ). QAT [35, 36] considers the quantization in the network training phase, while PTQ [4] quantizes the network after training. Due to its lower time consumption and computational resource requirements, PTQ is extensively employed in network deployment. Most of the work of PTQ involves learning the quantization parameters for weights and activations in each layer. In order to calculate the activations in the network, a small number of calibration samples should be used as input in PTQ. The selected quantization parameters are dependent with the selection of these calibration samples. [37, 5, 26] demonstrate the effect of the number of calibration samples. *Activation quantization is essentially a compression problem with a strong emphasis on maximizing the preservation of information contained in the activation*. Admittedly, heuristically designed distance metrics have achieved promising performance. However, we argue that previous works neglect the well-defined distributional metric in information theory, which is necessary for success measurement.

**Contrastive Learning and Mutual Information Maximization.** Recently, contrastive learning is proven to be an effective approach to MI maximization, and many methods based on contrastive loss for self-supervised learning are proposed, such as Deep InfoMax [19], Contrastive Predictive Coding [16], MemoryBank [15], Augmented Multiscale DIM [20], MoCo [21] and SimSaim [18]. These methods are generally rooted in NCE [17] and InfoNCE [19] which can serve as optimizing the lower bound of mutual information [38]. In the meantime, [22] and [39] generalize the contrastive idea into the content of knowledge distillation (KD) to pull-and-push the representations of teacher and student. Intuitively, the core principle of contrastive learning revolves around drawing representations from positive pairs closer, while pushing representations from negative pairs further apart in a contrastive space. One of the main challenges in employing contrastive loss is defining these negative and positive pairs.

Our approach for PTQ calibration harnesses the core concept of existing contrastive learning methods, notably contrastive-based network compression methods such as CRD [22], WCoRD [39], and MIM-BNN [23]. However, our methodology differs from these methods in several key ways: **(i)** The targeted mutual information (MI) we focus on and the numerical problem we formulate are entirely distinct; **(ii)** Our approach can naturally circumvent the cost of the MemoryBank [15] associated with the exponential number of negative pairs in related works, thanks to the limited calibration data size in our task; **(iii)** The discriminator functions we use are distinct. By leveraging the inherent characteristics of layer-wise quantization, where pre-trained full-precision network is available, we design a discriminator network specifically for PTQ, as discussed in Sec.3.1 and Appendix.4.

## 6 CONCLUSION

We argue that previous PTQ calibration works neglect the well-defined distributional metric in information theory, which is necessary for success measurement. To address this, we introduce a contrastive learning (CL) framework for PTQ calibration, focusing on maximizing the mutual information between pre- and post-quantization activations. This approach, which utilizes a self-supervised proxy task for optimizing quantization parameters, effectively retains more information in the quantized activations, enhancing the performance of the quantized model. Importantly, our method avoids common collusion, thanks to an ingeniously constructed critic function, ensuring effectiveness even under conditions of extremely limited calibration data.

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
