# A APPENDIX

## A.1 CODE IMPLEMENTATION

**Step 0: Codes can be found anomalously in here.**

## A.2 DETAILED PROOF OF THE EQUIVALENCE BETWEEN MINIMIZING CL-Calib LOSS IN EQ. 8 AND MAXIMIZING MUTUAL INFORMATION IN EQ. 16.

The key is to prove that obtaining $d^\star = \arg\min_d \mathcal{L}_{\texttt{CL-Calib}}$ by minimizing Eq.16 in main manuscript

$$
\begin{aligned}
\mathcal{L}_{\texttt{CL-Calib}} = \ &\mathbb{E}_{(\mathbf{A}_Q^k, \mathbf{A}_{F,+}^k)} \left[ -\log d(\mathbf{A}_Q^k, \mathbf{A}_{F,+}^k) \right] \\
&+ \mathbb{E}_{\{\mathbf{A}_{F,-,i}^k\}_{i=1}^M} \left[ -\log(1 - d(\mathbf{A}_Q^k, \mathbf{A}_{F,-,i}^k)) \right]
\end{aligned}
\tag{17}
$$

is equivalent to estimate the true distribution $P(D \mid \mathbf{A}_Q^k, \mathbf{A}_F^k)$, *i.e.,* , proving

$$
d^\star(\mathbf{A}_Q^k, \mathbf{A}_F^k) = P(D \mid \mathbf{A}_Q^k, \mathbf{A}_F^k).
\tag{18}
$$

$D$ is used to represent whether $(\mathbf{A}_Q^k, \mathbf{A}_F^k)$ is positive or negative (*i.e.,* , binary discrimination). Thus, distribution $P(D \mid \mathbf{A}_Q^k, \mathbf{A}_F^k)$ can be modeled as a Bernoulli distribution with

$$
d(\mathbf{A}_{Q,i}^k, \mathbf{A}_{F,j}^k) \triangleq \frac{\exp(g(\mathbf{A}_{Q,i}^k) \cdot g(\mathbf{A}_{F,j}^k)/\tau)}{\exp(g(\mathbf{A}_{Q,i}^k) \cdot g(\mathbf{A}_{F,j}^k)/\tau) + 1},
\tag{19}
$$

in which $d(\mathbf{A}_Q^k = \mathbf{A}_{Q,i}^k, \mathbf{A}_F^k = \mathbf{A}_{F,j}^k) \in [0,1]$. Without loss of generality, we can redefine the critic function $d$ as $d'$ such that $d'(D = 1, \mathbf{A}Q^k = \mathbf{A}Q, i^k, \mathbf{A}F^k = \mathbf{A}F, j^k) = d(\mathbf{A}Q^k = \mathbf{A}Q, i^k, \mathbf{A}F^k = \mathbf{A}F, j^k)$. It follows that $d'(D = 0, \mathbf{A}Q^k = \mathbf{A}Q, i^k, \mathbf{A}F^k = \mathbf{A}F, j^k) = 1 - d'(D = 1, \mathbf{A}Q^k = \mathbf{A}Q, i^k, \mathbf{A}F^k = \mathbf{A}F, j^k)$. The log likelihood is

$$
\mathbb{E}_{d \sim P(D \mid \mathbf{A}_Q^k = \mathbf{A}_{Q,i}^k, \mathbf{A}_F^k = \mathbf{A}_{F,j}^k)} \left[ \log d'(D = d, \mathbf{A}_Q^k = \mathbf{A}_{Q,i}^k, \mathbf{A}_F^k = \mathbf{A}_{F,j}^k) \right].
\tag{20}
$$

By Gibbs' inequality, the max likelihood fit is $d'(D = d, \mathbf{A}_Q^k = \mathbf{A}_{Q,i}^k, \mathbf{A}_F^k = \mathbf{A}_{F,j}^k) = P(D = d \mid \mathbf{A}_Q^k = \mathbf{A}_{Q,i}^k, \mathbf{A}_F^k = \mathbf{A}_{F,j}^k)$, which implies that $d(\mathbf{A}_Q^k = \mathbf{A}_{Q,i}^k, \mathbf{A}_F^k = \mathbf{A}_{F,j}^k) = P(D = 1 \mid \mathbf{A}_Q^k = \mathbf{A}_{Q,i}^k, \mathbf{A}_F^k = \mathbf{A}_{F,j}^k)$. By performing an expectation operation over $(\mathbf{A}_{Q,i}^k, \mathbf{A}F, j^k)$ on the aforementioned equation, we can derive the following expression:

$$
\mathbb{E}_{(\mathbf{A}_{Q,i}^k, \mathbf{A}_{F,j}^k) \sim P(\mathbf{A}_Q^k, \mathbf{A}_F^k)} \left[ \mathbb{E}_{d \sim P(D \mid \mathbf{A}_Q^k = \mathbf{A}_{Q,i}^k, \mathbf{A}_F^k = \mathbf{A}_{F,j}^k)} \left[ \log d'(D = d, \mathbf{A}_Q^k = \mathbf{A}_{Q,i}^k, \mathbf{A}_F^k = \mathbf{A}_{F,j}^k) \right] \right]
\tag{21}
$$

$$
= \mathbb{E}_{d, (\mathbf{A}_{Q,i}^k, \mathbf{A}_{F,j}^k) \sim P(D, \mathbf{A}_Q^k, \mathbf{A}_F^k)} \left[ \log d'(D = d, \mathbf{A}_Q^k = \mathbf{A}_{Q,i}^k, \mathbf{A}_F^k = \mathbf{A}_{F,j}^k) \right]
\tag{22}
$$

$$
= \mathbb{E}_{(\mathbf{A}_{Q,i}^k, \mathbf{A}_{F,j}^k) \sim P(\mathbf{A}_Q^k, \mathbf{A}_F^k \mid D=1) P(D=1)} \left[ \log d(\mathbf{A}_Q^k = \mathbf{A}_{Q,i}^k, \mathbf{A}_F^k = \mathbf{A}_{F,j}^k) \right]
\tag{23}
$$

$$
+ \mathbb{E}_{(\mathbf{A}_{Q,i}^k, \mathbf{A}_{F,j}^k) \sim P(\mathbf{A}_Q^k, \mathbf{A}_F^k \mid D=0) P(D=0)} \left[ \log(1 - d(\mathbf{A}_Q^k = \mathbf{A}_{Q,i}^k, \mathbf{A}_F^k = \mathbf{A}_{F,j}^k)) \right]
\tag{24}
$$

$$
= \mathbb{E}_{(\mathbf{A}_Q^k, \mathbf{A}_{F,+}^k)} \left[ \log d(\mathbf{A}_Q^k, \mathbf{A}_{F,+}^k) \right] + \mathbb{E}_{\{\mathbf{A}_{F,-,i}^k\}_{i=1}^M} \left[ \log(1 - d(\mathbf{A}_Q^k, \mathbf{A}_{F,-,i}^k)) \right],
\tag{25}
$$

which is the negative of Eq.8. Therefore, minimizing CL-Calib Loss in Eq. 8 and maximizing mutual information in Eq. 16. Note that the proof is based on the proof in [22] and [16].

## A.3 MORE EXPERIMENTS WITH TRANSFORMER ARCHITECTURES.

Similarly to PD-Quant [6], we also extend our proposed method to Transformer models. Our method is evaluated on both Vision Transformer (ViT) [33] and Data-efficient Image Transformer (DeiT) [41] across different bit settings. The results demonstrate the broad generalization ability of our proposed CL-Calib method.

Table 3: Comparison with state-of-the-art PTQ methods with Transformer models. ViTS/16/224 denotes patch size is $16 \times 16$, the input resolution is $224 \times 224$. All the results listed are the top-1 accuracy (%).

| Model | Method | Bits (W/A) | Acc(%) |
|---|---|---|---|
| ViT-S/16/224 74.65% | PTQ4ViT [40] | W6A6 | 70.72 |
| | QDrop [10] | | 70.25 |
| | PD-Quant [40] | | 70.84 |
| | CL-Calib (ours) | | **71.56** |
| | PTQ4ViT [40] | W4A6 | 53.55 |
| | QDrop [10] | | 67.57 |
| | PD-Quant [40] | | 68.64 |
| | CL-Calib | | **68.68** |
| | PTQ4ViT [40] | W2A6 | 0.31 |
| | QDrop [10] | | 45.16 |
| | PD-Quant [40] | | 48.13 |
| | CL-Calib (ours) | | **49.73** |
| DeiT-S/16/224 79.71% | PTQ4ViT [40] | W6A6 | 76.83 |
| | QDrop [10] | | 77.95 |
| | PD-Quant [40] | | 78.33 |
| | CL-Calib (ours) | | **79.09** |
| | PTQ4ViT [40] | W4A6 | 74.17 |
| | QDrop [10] | | 77.66 |
| | PD-Quant [40] | | 77.88 |
| | CL-Calib (ours) | | **78.50** |
| | PTQ4ViT [40] | W2A6 | 3.79 |
| | QDrop [10] | | 65.76 |
| | PD-Quant [40] | | 67.53 |
| | CL-Calib (ours) | | **67.99** |

## A.4 HYPER-PARAMETERS AND RELATIVE MODULE SELECTION.

In addition to ablation studies, we conduct experiments to select the important hyper-parameters and modules. We investigate two hyper-parameters in the CL-Calib loss in Eq.8 (especially the hyper-parameters in discriminator function $d$), the co-efficient $\tau$ to adjust the temperature in the CL-Calib loss in Eq.8, the number of negative samples $M$, and the architecture of embedding network $g$ in critic function.

**Number of negative samples $M$ in CL-Calib Loss.**

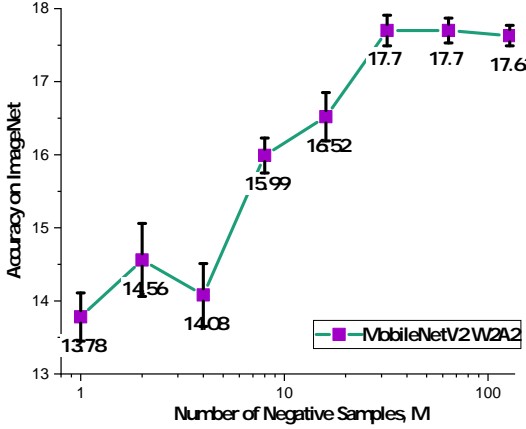

Figure 7: **Effect of Number of Negative Samples $M$.** When $M \geq 32$, the performance improvement reaches a plateau, which suggests that 32 negative samples are sufficient for CL-Calib.

Theoretically, in the context of contrastive learning, a larger value of $M$ in the `CL-Calib` Loss (as per Equation 8) results in a tighter lower bound on the Mutual Information. To circumvent the necessity of using an exceedingly large batch size, Wu et al. [15] were the first to propose the implementation of a memory buffer. This buffer stores the latent features of each data sample, computed from preceding batches. Inspired by [15], during the training process, many contrastive learning works [42, 22, 16] efficiently fetch a considerable number of negative samples from this memory buffer. It becomes a common technique for contrastive learning with a huge number of negative samples (*e.g.,* , Tian *et al.* [22] use 16,384 negative samples per instance for ImageNet classification). However, due to the setting of post-training quantization, the number of calibration samples cannot be huge. Combining the formulation for negative samples in Sec.3.1, we deduce that the maximum number of negative samples in `CL-Calib` is equivalent to the size of calibration samples (which is 128 for the experiments in Tab.1 in the main manuscript). To investigate the influence of $M$, we perform a sequence of experiments using MobileNetV2 in a W2A2 configuration. The results are shown in Fig. 7. We can witness that when $M \geq 32$, the number of negative samples is enough for `CL-Calib`.

**Temperature $\tau$ `CL-Calib` Loss.**

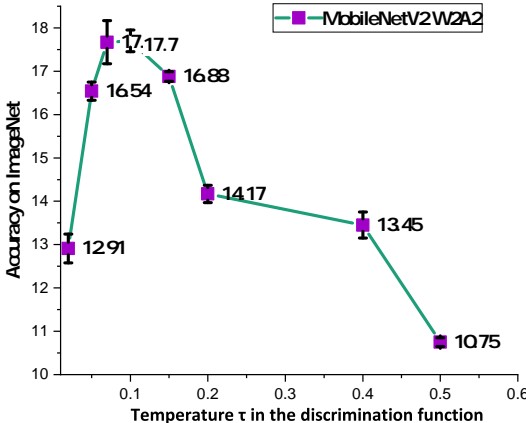

Figure 8: **Effect of Temperature $\tau$.** When approximately $\tau = 0.1$, the performance reaches a plateau.

To investigate the influence of $\tau$, we similarly perform a sequence of experiments using MobileNetV2 in a W2A2 configuration. The results are shown in Fig. 8. We observe that the performance peaks when $\tau = 0.1$, thus we adopt $\tau = 0.1$ as the default setting in our method.

**Architecture of mapping network $g$ in discrimination function $d$.**

Table 4: Architecture selection with MobileNetV2 [30] PTQ setting.

| Architecture of $g$ | Accuracy |
|---|---|
| no network | 12.68 |
| with 1 fc-layer | 12.76 |
| with 2 fc-layers | 12.34 |
| with 1 conv-layer | 12.52 |
| with 2 conv-layers | 12.51 |
| with 1 conv-fc-module | 13.12 |
| with 2 conv-fc-module | 12.94 |
| with 1 conv-fc-bn-module | 13.09 |
| with 2 conv-fc-bn-module | 13.09 |
| with learnable $f_F^{[k:]}$ | 11.08 |
| with fixed $f_F^{[k:]}$ | **17.70** |

To investigate the influence of architectures of discriminator function, we perform a sequence of experiments using MobileNetV2 in a W2A2 configuration. And to optimize $g$, we utilize the Adam optimizer with a learning rate of 0.001, in conjunction with a cosine learning rate scheduler. The results are shown in Fig. 4. We note that employing the fixed $f_F^{[k:]}$ as the embedding network $g$ in the discriminator function $d$ significantly outperforms other network architectures that require optimization.

We attempt to understand this occurrence through the lens of representation collapse in the context of contrastive learning. Although the objective of combined self-supervised techniques is to acquire significant representations, a considerable proportion of these strategies grapple with what's known as dimensional collapse. Dimensional collapse occurs when information encoded across different dimensions of the representation is redundant [43, 44, 45, 46]. In summary, the contrastive space requires thoughtful design and the mapping function demands meticulous optimization. Fortunately, we have access to the pre-trained full-precision model, making it straightforward to map quantized activations into the prediction space using this pre-trained model. In this way, we can avoid the collapse solution often encountered in contrastive learning as shown in the above table.

**Similarity Metrics Selection.**

We evaluated L1, L2, MSE, inner dot product, and cosine similarity as our similarity metrics. The findings, as depicted in Table 6, indicate that the inner dot product has a slight edge over the other metrics in our context.

### A.5    SHARED MUTUAL INFORMATION

We have adapted the core ideas and code from [47, 48, 49] to visualize the MI between quantized and full-precision activations. Results presented in Fig. 9 distinctly illustrate that the desired MI is effectively maximized post-training with CL-Calib loss.

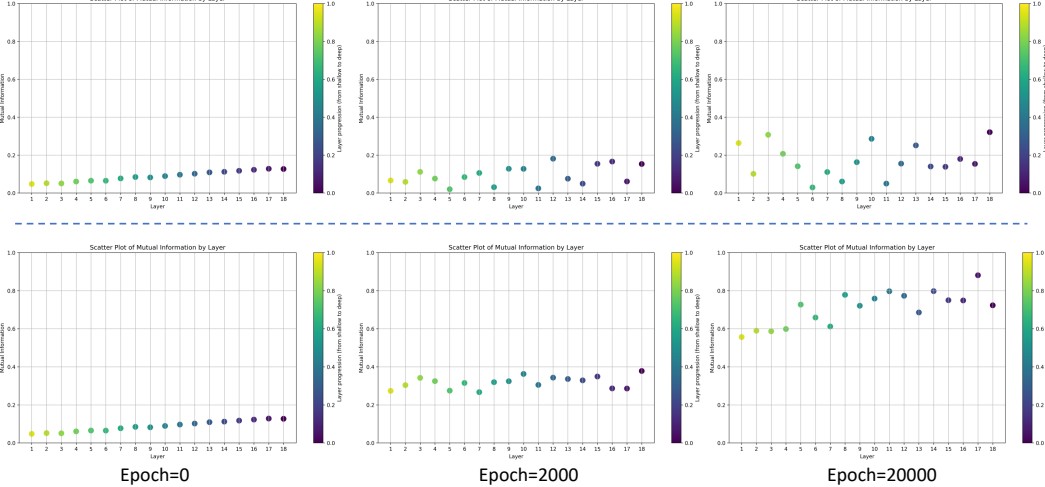

Figure 9: **Mutual Information between Full-precision and Quantized Activations, $I(\mathbf{A}_F^l, \mathbf{A}_Q^l)$:** Using ResNet18 in a 2W2A configuration as an example, the top plot showcases the changes in mutual information during training without CL-Calib. In contrast, the bottom plot illustrates the changes when training with CL-Calib. It's evident that utilizing CL-Calib for calibration enhances the mutual information between full-precision and quantized activations.

### A.6    RESULTS ON OBJECT DETECTION

We conduct experiments on object detection using the MS-COCO dataset, in a manner consistent with Qdrop [10]. Both two-stage (Faster RCNN) and one-stage (RetinaNet) models were employed, with backbones ranging from ResNet-18 and ResNet-50 to MobileNetV2. Our quantization strategy ensured that the first and last layers remain at 8-bit precision, while the neck (FPN) of the model is

Table 5: Comparisons of typical post-training quantization strategies in terms of mAP on MS COCO for object detection.

| Method | Bits (W/A) | Faster RCNN | | | RetinaNet | | |
|---|---|---|---|---|---|---|---|
| | | ResNet-18 | ResNet-50 | MobileNetV2 | ResNet-18 | ResNet-50 | MobileNetV2 |
| Full Prec. | 32/32 | 34.60 | 38.56 | 33.47 | 33.22 | 36.80 | 32.63 |
| AdaRound | 4/4 | 32.57 | 34.47 | 26.11 | 31.04 | 33.51 | 24.99 |
| BRECQ | 4/4 | 32.58 | 34.59 | 26.58 | 31.21 | 33.47 | 24.84 |
| QDROP | 4/4 | 33.37 | 36.96 | 30.88 | 31.99 | **35.67** | 29.75 |
| QDROP+CL-Calib | 4/4 | **33.71** | **37.22** | **31.54** | **32.34** | 35.61 | **30.28** |
| BRECQ | 2/4 | 29.92 | 30.23 | 19.35 | 28.73 | 29.47 | 18.46 |
| QDROP | 2/4 | 31.01 | 34.23 | 25.04 | 29.69 | 33.01 | 24.89 |
| QDROP+CL-Calib | 2/4 | **31.10** | **35.03** | **26.35** | **30.24** | **33.58** | **25.21** |

Table 6: Selecting similarity metric in contrastive learning with MobileNetV2 PTQ setting.

| similarity metric | Accuracy |
|---|---|
| L1 | 9.56 |
| L2 | 16.22 |
| MSE | 16.13 |
| cosine similarity | 17.34 |
| inner dot product | **17.70** |

Table 7: Overfitting problem verification: Train and test accuracy on ResNet-18 W2A2.

| Method | Train Accuracy | Test Accuracy |
|---|---|---|
| w.o CL-Calib | 68.93 | 51.28 |
| w. CL-Calib | 65.31 | **54.45** |

quantized. As with our transformer experiments, the results from these experiments (refer to Table 5) indicate the robustness and generalizability of CL-Calib, even without an extensive hyper-parameter search.

### A.7 Overfitting in calibration

We aimed to compare the performance accuracy between the calibration data and test data. Taking ResNet-18 W2A2 as an example, our results (presented in Table 7) demonstrate that, especially under extremely low-bit quantization, calibration without the incorporation of CL-Calib shows favorable performance on the calibration data, but a noticeable drop in accuracy on the test data. This disparity between the performances is a clear indication of overfitting when conducting layer-wise calibration.

## B Related Work

Quantization stands as one of the most potent techniques for compressing neural networks. It can be broadly categorized into two primary methods: Quantization-Aware Training (QAT) and Post-Training Quantization (PTQ). QAT [35, 36] considers the quantization in the network training phase, while PTQ [4] quantizes the network after training. Due to its lower time consumption and computational resource requirements, PTQ is extensively employed in network deployment. Most of the work of PTQ involves learning the quantization parameters for weights and activations in each layer. Take uniform quantization as an example, the quantization parameters include scaling factor $S$ and zero point $z$. The floating-point value $x$ is quantized to integer value $x_{int}$ in the form of: $x_{int} = \text{clamp}(\lfloor \frac{x}{s} \rceil - z, p_{min}, p_{max})$. The clamp function clip the rounded value $\lfloor \frac{x}{s} \rceil - z$ to the range of $[p_{min}, p_{max}]$. In order to set quantization parameters for the weight tensor and the activation tensor in a layer, a simple but effective way is to select the quantization parameters that minimize the MSE of the tensors before and after quantization [50, 9, 11, 37]. Other metrics, such as L1 distance, cosine distance, and KL divergence, can also be used to evaluate the distance of the tensors before and after quantization [51, 40].

In order to calculate the activations in the network, a small number of calibration samples should be used as input in PTQ. The selected quantization parameters are dependent with the selection of these calibration samples. [37, 5, 26] demonstrate the effect of the number of calibration samples. *Activation quantization is essentially a compression problem with a strong emphasis on maximizing the preservation of information contained in the activation.* Admittedly, heuristically designed distance metrics have achieved promising performance. However, we argue that previous works neglect the well-defined distributional metric in information theory, which is necessary for success measurement.

## B.1 CONTRASTIVE LEARNING AND MUTUAL INFORMATION MAXIMIZATION

Recently, contrastive learning is proven to be an effective approach to MI maximization, and many methods based on contrastive loss for self-supervised learning are proposed, such as Deep Info-Max [19], Contrastive Predictive Coding [16], MemoryBank [15], Augmented Multiscale DIM [20], MoCo [21] and SimSaim [18]. These methods are generally rooted in NCE [17] and InfoNCE [19] which can serve as optimizing the lower bound of mutual information [38]. In the meantime, [22] and [39] generalize the contrastive idea into the content of knowledge distillation (KD) to pull-and-push the representations of teacher and student. Intuitively, the core principle of contrastive learning revolves around drawing representations from positive pairs closer, while pushing representations from negative pairs further apart in a contrastive space. One of the main challenges in employing contrastive loss is defining these negative and positive pairs. In our approach, we utilize contrastive learning to maximize our targeted mutual information (MI).

Our approach for PTQ calibration harnesses the core concept of existing contrastive learning methods, notably contrastive-based network compression methods such as CRD [22], WCoRD [39], and MIM-BNN [23]. However, our methodology differs from these methods in several key ways: **(i)** The targeted mutual information (MI) we focus on and the numerical problem we formulate are entirely distinct; **(ii)** Our approach can naturally circumvent the cost of the MemoryBank [15] associated with the exponential number of negative pairs in related works, thanks to the limited calibration data size in our task; **(iii)** The discriminator functions we use are distinct. By leveraging the inherent characteristics of layer-wise quantization, where pre-trained full-precision network is available, we design a discriminator network specifically for PTQ, as discussed in Sec.3.1 and Appendix.4.