# OpenReview forum: "CL-Calib: Enhancing Post-training Quantization Calibration through Contrastive Learning"
_ICLR.cc/2024/Conference — ICLR 2024 Conference Withdrawn Submission_

### Official Review · Reviewer_Wd4w · 2023-10-29

**Soundness:** 3 good
**Presentation:** 1 poor
**Contribution:** 2 fair
**Rating:** 3
**Confidence:** 3

**Summary:**

This paper proposes CL-Calib, which is a contrastive learning-based activation calibration method for post-training quantization (PTQ). The core insight of CL-Calib is to formulate the parameter calibration as a self-supervised proxy task and maximize the mutual information between FP and post-quantized activation. The author theoretically proves that minimizing the proposed loss is the same as maximizing the mutual information. The proposed scheme is verified on various network architectures on the ImageNet dataset.

**Strengths:**

1. The theoretical analysis and proof for the mutual information (Sec 3.2) are comprehensive and insightful.

2. Many network architectures are evaluated with the proposed CL-Calib in the experiments, which improves the soundness of this work.

**Weaknesses:**

1. The presentation quality needs to be improved, including but not limited to:
* Figure 1 is misleading, Why FP activation is extracted from the intermediate layer of the quantized network but the quantized activation is not? Why backward pass does not flow through the quantized network?
* Many typos need to be fixed, e.g. Sec.1 contrastive learning (CT) => (CL), (e.g., ,) => (e.g.,) (remove one comma). Figure 1 Quantize Parameter => Quantization Parameter. Sec.2 (i.e., ,) => (i.e.,) (remove one comma), Eq. 6 there is one mismatch "}", etc.
* The first subsection of section 3.1 (Instance Recognition (IR) for PTQ Calibration) has nearly no relationship with quantization, I suggest the author move this part to section 2 as the preliminary of contrastive learning.

2. The motivation of this work is not very clear and straightforward. It seems more like a combination of contrastive learning and calibration of PTQ. I fully understand that the proxy contrastive learning task is the same as the optimization problem of Eq.5 as proved in Sec 3.2, but what is the real and core advantage of using this proxy CL task instead of directly optimizing the target?

3. More tasks of PTQ other than image classification should be verified, such as object detection and even some NLP tasks. Also, experiments on Transformer-based model (including ViTs) is a plus. In Table 1, the author mentions that all experiments are implemented by them but why some results of specific methods on specific bit-width or models are missing? (ACIQ-Mix and Bit-split are not evaluated on RegNetX and MNasx2, BRECQ is not evaluated on 4/2, 2/4-bit, etc.)

**Questions:**

Please see the weakness above.

---

### Official Review · Reviewer_CeZB · 2023-10-30

**Soundness:** 2 fair
**Presentation:** 2 fair
**Contribution:** 2 fair
**Rating:** 5
**Confidence:** 4

**Summary:**

In this paper, a self-supervised proxy task is proposed to assist in post-training quantization (PTQ) by using contrastive learning, named as CL-Calib. The main contribution of this work is that the CL-Calib can be an "add-on module" for state-of-the-art PTQ methods to help increase quantization accuracy.

**Strengths:**

1. this work use a small amount of data (for example, 128 images in ImageNet) to construct positive and negative sample pairs for adjusting quantization parameters, i.e., scaling factor and zero point, for activations by using contrastive learning. This idea is incrementally novel.
2. This module can be incorporated into many PTQ framework to improve accuracy.

**Weaknesses:**

1. The  idea of contrastive learning for tuning quantization parameters is straightforward but also need very careful design of selecting positive and negative samples in different down-streaming tasks. In this paper, this design for tasks such as object detection is not clearly stated and the experimental setting is also not very clear. This makes this work not sounded for these tasks.
2. The training stability of contrastive learning under such limited training images is mentioned but not very fully explained in this paper. How and why the mapping function can solve that problem very well theoretically and experimentally. In this paper, the experimental results did not provide a convincing evidence that this problem can be easily solved. For example, all experimental results should be displayed as a stable performance improvement under different selection of training samples rather than the best result shown in all tables and figures.

**Questions:**

1. Please provide more information about the issues in the weakness section.
2. Compared with QAT, we care about the efficiency of PTQ. So, how about the training time for CL-Calib for image classification and object detection?
3. How about the performance of CL-Calib on tasks such as regression, image segmentation,  and other downstreaming tasks?
4. If we use only L_CL-Calib without L_quant, the training process works or not, and why?

---

### Official Review · Reviewer_Rw23 · 2023-11-01

**Soundness:** 3 good
**Presentation:** 2 fair
**Contribution:** 3 good
**Rating:** 6
**Confidence:** 5

**Summary:**

The author proposes a contrastive learning based framework to learn the quantization parameters of activations. A theoretical guarantee is presented to illustrate minimizing the designed loss is equivalent to maximizing the desired mutual information.

**Strengths:**

CL is an interesting idea in PTQ. Good performance.

**Weaknesses:**

1, This paper does not provide the overall process. It only optimizes the quantization parameters of activations. Is that all? Nothing else? It confused me. A more clear description is needed.
2, There is some confusion when Eq15 comes to Eq16. The author says: we can further loosen the lower bound to obtain the optimization objective. However, it is more likely a formula that's been forcibly pieced together.
3, There is some missing paper that also uses contrastive learning in PTQ. [1]
4, I do believe the proof in Appendix A.2 has some typos. For example, symbol between Eq15 and Eq16.

[1] CPT-V: A Contrastive Approach to Post-Training Quantization of Vision Transformers

**Questions:**

As the author claims, QDrop, BRECQ, and PD-Quant all focus on activation calibration. However, as far as I know, BRECQ and PD-Quant rarely discuss activation calibration.

'in this work We only consider uniform unsigned symmetric quantization': We -> we


I am willing to accept this paper at this time and the final rating is upon the rebuttal.

---

### Official Review · Reviewer_ZHZD · 2023-11-01

**Soundness:** 3 good
**Presentation:** 3 good
**Contribution:** 2 fair
**Rating:** 5
**Confidence:** 1

**Summary:**

This study introduces a approach to Post-Training Quantization (PTQ) calibration by maximizing mutual information between pre- and post-quantized activations. A Contrastive Learning (CL) framework is proposed for calibration, optimizing quantization parameters through a self-supervised proxy task. By incorporating CL, the method effectively utilizes positive pairs of quantized and Floating-Point (FP) activations from the same input samples and distinguishes them from negative pairs from different samples. The approach addresses common issues in CL, especially in limited calibration data scenarios, thanks to a well-designed critic function. The study also provides a theoretical guarantee, demonstrating that minimizing the proposed loss is equivalent to maximizing the desired mutual information. Experimental results indicate the method's effectiveness as an additional module for state-of-the-art PTQ methods, highlighting its practical utility and efficiency in enhancing quantized network performance.

**Strengths:**

CL-Calib stands out due to its integration of self-supervised learning and contrastive learning within PTQ calibration. By maximizing mutual information between quantized and full-precision activations, it provides a deep theoretical understanding of the quantization process. CL-Calib's versatility is showcased through seamless integration with state-of-the-art PTQ methods, making it a powerful and adaptable tool for enhancing quantized neural network performance.

**Weaknesses:**

1. The improvement in performance with 4-bit quantization is not substantial, and the absence of experiments in 8-bit quantization and language model domains limits the breadth and depth of the study.
2. The study lacks significant innovation, as it does not introduce novel methods or techniques that significantly differ from existing research, making its innovative contribution less prominent.
3. The practical utility of the proposed approach is constrained. Despite achieving results in 2-bit quantization, the performance gap compared to full-precision models remains considerable, restricting its applicability in real-world scenarios.

**Questions:**

1. Is it possible to provide a clear description of the baseline performance in the experimental table? Specifically, does the table indicate whether the baseline results were achieved by combining QDrop and PD-quant, or were these methods used individually?

2.It appears that there is no comparison made with the results presented in the paper available at https://arxiv.org/pdf/2208.11945.pdf. There seems to be a noticeable gap, especially in the 4-bit quantization results. Could you please address this discrepancy?